# Therapeutic Options in Postoperative Enterocutaneous Fistula—A Retrospective Case Series

**DOI:** 10.3390/medicina58070880

**Published:** 2022-06-30

**Authors:** Maria Mădălina Denicu, Dan Cartu, Mihai Ciorbagiu, Raducu Nicolae Nemes, Valeriu Surlin, Sandu Ramboiu, Luminița Cristina Chiuțu

**Affiliations:** 1I.C.U. Clinic, Clinical County Emergency Hospital of Craiova, 1. Tabaci Street, 200642 Craiova, Romania; magda_petre@yahoo.es (M.M.D.); luminita.chiutu@gmail.com (L.C.C.); 2Doctoral School, Faculty of Medicine, University of Medicine and Pharmacy of Craiova, 2-4 Petru Rares Street, 200349 Craiova, Romania; raducunemes@yahoo.com; 31st Clinic of Surgery, Clinical County Emergency Hospital of Craiova, 1. Tabaci Street, 200642 Craiova, Romania; cartu_dan@hotmail.com (D.C.); vsurlin@gmail.com (V.S.); 46th Department, Faculty of Medicine, University of Medicine and Pharmacy of Craiova, 2-4 Petru Rares Street, 200349 Craiova, Romania; 52nd Clinic of Surgery, Clinical County Emergency Hospital of Craiova, 1. Tabaci Street, 200642 Craiova, Romania; mihai.ciorbagiu@gmail.com; 67th Department, Faculty of Medicine, University of Medicine and Pharmacy of Craiova, 2-4 Petru Rares Street, 200349 Craiova, Romania

**Keywords:** entero-cutaneous fistula, multidisciplinary approach, conservative treatment

## Abstract

Objectives: The aim of the study was to present the results obtained in our experiment regarding the management of postoperative enterocutaneous fistulas (PECF). Materials and Methods: We conducted a retrospective study on 64 PECF registered after 2030 abdominal surgeries (1525 digestive tract surgeries and 505 extra-digestive ones) over a period of 7 years (1st of January 2014–31th of December 2020) in the 1st and 2nd Surgery Clinics, Clinical County Emergency Hospital of Craiova, Romania. The group included 41 men (64.06%) and 23 women (35.34%), aged between 21–94 years. Of the cases, 71.85% occurred in elderly patients over 65 years old. Spontaneous fistulas in Crohn’s disease, intestinal diverticulosis, or specific inflammatory bowel disease were excluded. Results: The overall incidence of 3.15% varied according to the surgery type: 6.22% after gastroduodenal surgery, 1.78% after enterectomies, 4.30% after colorectal surgery, 4.28% after bilio-digestive anastomoses, and 0.39% after extra-digestive surgery. We recorded a 70.31% fistula closure rate, 78.94% after exclusive conservative treatment and 57.61% after surgery; morbidity was 79.68%, mortality was 29.68%. Conclusion: PECF management requires a multidisciplinary approach and is carried out according to an algorithm underlying well-established objectives and priorities. Conservative treatment including resuscitation, sepsis control, output control, skin protection, and nutritional support is the first line treatment; surgery is reserved for complications or permanent repair of fistulas that do not close under conservative treatment. The therapeutic strategy is adapted to topography, morphological characteristics and fistula output, age, general condition, and response to therapy.

## 1. Introduction

Postoperative enterocutaneous fistulas (PECF), one of the most severe and dramatic complications of abdominal surgery, are discouraging for the surgeon, disabling for the patient, and are still burdened by a constant increase in morbidity rates (85–90%), significant mortality (5–20%) [1,2,3,4,5,6], prolonged hospitalization, and high costs, despite substantial therapeutic advances over the past 40 years.

Using a wide range of conservative and surgical therapeutic means, the treatment of PECF aims to close the fistula and restore the digestive transit, with the lowest morbidity and mortality. Such objectives require a multidisciplinary approach with a complex team (surgeon, anesthetist, internist, radiologist, nutritionist, nurses specialized in bedside care, wound care and stoma therapy, psychotherapist, social worker, etc.).

The basic principles established by Chapman in 1964 [7] have been supplemented with new techniques and methods. Currently, there is a consensus on PECF management and a multistage algorithm. Each stage involves well-established goals and priorities: fistula identification, stabilization (resuscitation), sepsis control, output control, skin protection, definition of fistula morphology, nutritional support, and definitive treatment (surgery) [8].

We performed a retrospective study upon the management PECF cases encountered in the experience of two surgical departments in our hospital to evaluate the results we obtained using modern principles of management adapted to the specific etiology and morphologic features.

## 2. Materials and Methods

This retrospective study was conducted on an aggregated group of 64 PECF patients registered after 2030 abdominal surgeries (1525 digestive tract surgeries and 505 extra-digestive ones) performed in the 1st and the 2nd Surgery Clinics of the Clinical County Emergency Hospital of Craiova over seven years (2014–2020) and aimed at evaluating the therapeutic strategies. The group included 41 men (64.06%) and 23 women (35.34%), aged 21–94 years; 71.85% of the cases occurred in elderly patients over 65 years old. Spontaneous fistulas in Crohn’s disease, intestinal diverticulosis, or specific inflammatory bowel diseases were excluded. Demographic data, data about primary intervention, patient’s co-morbidities, diagnostic circumstances of postoperative fistula, laboratory test results, imaging investigations, surgical re-interventions’ findings, applied therapeutic measures, nutrition, nasogastric suction, drainage, output of fistula, wound care, and skin care were extracted from the clinical observation files, surgery transcripts, anesthesia, intensive care files, imaging investigation, and autopsy reports. The study was approved by the Craiova Emergency Clinical Hospital’s Ethics Committee.

## 3. Results

Overall incidence of enterocutaneous fistula was 3.15%; 6.22% after gastroduodenal surgery, 1.78% after enterectomies, 4.30% after colorectal surgery, 4.28% after biliodigestive anastomoses, and 0.39% after extra-digestive surgery.

The primary lesion requiring surgery was malignant in 54 (84.37%) cases (gastric cancer–16, colorectal cancer–31, pancreatic head cancer–2, vaterian ampuloma–1, small bowel–2, retroperitoneal tumor–1, recurrent gastric cancer–1) and benign in 10 cases (bleeding gastroduodenal ulcer, sigmoid volvulus, recto-colic polyposis, acute lithiasic cholecystitis, entero-mesenteric ischemia, adherences/perivisceritis, strangulated parastomal hernia, strangulated postoperative incisional hernia, subphrenic abscess after operated liver hydatid cyst). There were 38 elective surgery cases, 26 (40.62%) of which the following were emergencies: intestinal occlusion–16 cases, peritonitis–5 cases, upper digestive bleeding–3 cases, lower digestive bleeding–1 case, and acute abdomen of unspecified cause–1 case. The type of surgery and the primary anastomosis were adapted to the patient’s intraoperative lesion findings, age, and biological condition.

The onset of the fistula was early (in the first 2–7 postoperative days) in 27 (42.18%) of the cases and late (over seven postoperative days) in 37 (57.82%) of the cases. The diagnosis was established clinically in most cases, with the onset of purulent discharge or intestinal contents through the wound, on the drain tubes, or both, confirming the diagnosis as occurring in 67.18% of the cases, preceded or accompanied by localized or generalized abdominal pain, general symptoms of sepsis, peritoneal or occlusive syndrome, increased volume of upper digestive aspiration, or the need to reinstall it; in cases with uncertain diagnosis (8 cases), the diagnosis was confirmed by the methylene blue test.

Imaging investigations (fistulography–9 cases, CT-scan–22 cases, and intestinal follow-through with water-soluble contrast–14 cases), performed after the stabilization in the patient (7–10 days), allowed defining the morphological characteristics of the fistula, as follows: simple, with direct tract–30 cases, complex–34 cases (complex or multiple tracts–32 cases, and entero-atmospheric fistula–2 cases). The origin was: esophageal–10 cases, gastric–7, duodeno-biliary–6, colorectal–24, pancreas–2, and bladder–1. Skin openings were single in 44 cases (operative wound–16 or drains–28) and multiple in 18 cases (wound + drains); 2 cases were enteroatmospheric fistulas.

The fistula output was low (<200 mL/day) in 27 cases, medium (200–500 mL/day) in 21 cases, and high (>500 mL/day) in 16 cases.

Treatment was started immediately after diagnosis. Conservative medical treatment was the first-line treatment in all cases. It was performed according to an algorithm that included the following objectives (Table 1): patient stabilization (resuscitation), sepsis control, output control, nutrition, wound care, and skin protection. Of the total cases, 38 (56.25%) were treated exclusively and conservatively. Volemic, hydro-electrolytic, acid-base, and hematologic resuscitation took place in the intensive care unit (ICU) in 35 cases (59.37%).

Of the total patients, 26 (43.75%) had surgery and 9 cases in the first 7–10 days postoperatively. Indications for early operations were postoperative peritonitis, intraperitoneal abscesses confirmed by CT-scan, unfavorable evolution under conservative treatment, or increased output over 1000 mL/day (Table 2).

The surgical procedures we used were adapted to the intraoperative findings. Early interventions addressed the septic complications (abscess, acute peritonitis) or obstruction. Late interventions aimed at removing the fistula and restoring the transit (Table 3).

The fistula closure rate was 70.31% (45 cases): 78.94% in those treated exclusively and conservatively and 57.61% in those treated conservatively and surgically. Fifty-one complications (29 general and 30 local) were registered, with a morbidity rate of 79.68%. The overall mortality rate was 29.68%, the leading causes of death being sepsis, malnutrition, hemorrhagic shock, respiratory complications, and pulmonary thromboembolism. (Table 4) There were no recurrences of fistula after closure both in the conservative and conservative plus surgery groups of patients.

## 4. Discussion

The treatment of PECF is difficult, complex, and long-lasting, aiming to close the fistula and restore the digestive tract with minimal morbidity, and it requires the multidisciplinary approach of an integrated team.

Stabilization (resuscitation) of the patient is, along with sepsis control, the absolute priority in the management of patients with PECF, given that they are hypovolemic, dehydrated, and with severe electrolyte imbalances from the beginning due to fistula losses and the formation of the third sector of fluid retention secondary to peritoneal infection [1,5,6,9,10,11,12,13]. The degree of fluid depletion varied between 50 and 3000 mL/day depending on the topography of the fistula, the most common abnormalities being hypovolemia, hypokalemia, and metabolic acidosis [14]. Resuscitation was started in the first 24–48 h after diagnosis, with the objectives of assessing losses, aggressive fluid resuscitation, correcting electrolyte imbalances, and acidosis. In 35 cases (59.37%), there were fistulas with high output, significant hypovolemia, and major electrolytic imbalances. Resuscitation took place in the ICU for a variable period (average of 9.33 days, with a minimum of two and a maximum of 22 days), depending on the severity and therapeutic response. [1,12,15,16].

Leakage assessment, essential for determining fluid and electrolyte requirements, involved accurate output measurement, correlated with size, topography, effluent content and composition, serum electrolyte level, and other balance parameters (such as urine output, number, and weight of dressings) [16,17,18]. The assessment of the fluid balance and monitoring of the input/output balance must be corrected frequently (every 4–8 h) until the patient’s stabilization, especially in high-output fistula patients, which are highly vulnerable to significant electrolyte imbalances due to loss of sodium, potassium, chlorine, and bicarbonate, and malnutrition, tending to progress toward MSOF (multisystemic organic failure) and being burdened by increased morbidity and mortality [1,6,10,15]. Loss assessment is sometimes difficult, and in patients with peritonitis, resuscitation cannot be complete as long as peritoneal contamination is not controlled; therefore, resuscitation must be continued intraoperatively [12].

The correction of volume and electrolyte imbalances was carried out with crystalloid solutions, to which we added electrolytes depending on the balance; hypokalemia, the most common electrolyte imbalance, was corrected immediately (10 mEq KCl/liter infusion solution) to prevent cardiac complications (arrhythmias), organ damage, and death associated with hypokalemia [6,19]. In duodenal fistulas with a high output rate, the loss of pancreatic secretions requires the addition of bicarbonate. In intestinal fistulas with a high output rate and long evolution, it was necessary to supplement with zinc, vitamins, and micro-elements in doses up to 10 times the average amount. To correct anemia, 32 (50%) patients were transfused with whole blood, improving the oxygen transport capacity, while albumin infusion restored the oncotic plasma pressure [9].

Sepsis control. Sepsis is the leading cause of death in PECF, estimated by some authors to be as high as 77% [2,6,8,9,10,11,12,15,20]. Therefore, sepsis control is a paramount objective, carried out simultaneously with the patient’s stabilization, and started immediately when the patient is stable enough to support the diagnostic and therapeutic procedures [2,6,10,21]. The management of sepsis involves identifying the source, drainage, and antibiotic therapy. Computed tomography is the best tool for diagnosing intra-abdominal collections or free fluid in the peritoneal cavity and guides percutaneous abscess drainage [1,2,8,9,10,11,12,14,17,21,22]. In addition, the injection of a water-soluble contrast agent after evacuation allows obtaining the data about the internal orifice, the fistulous path, the continuity of the intestine, and the condition of the adjacent loops. In patients with severe sepsis without response to resuscitation, significant anastomotic dehiscence and generalized peritonitis, or abscesses that cannot be drained percutaneously, sepsis control requires an emergency reintervention for evacuation, peritoneal drainage, and fistula control by externalizing the intestinal loop with fistula, diversion, or proximal stoma [1,8,10,12,21]. We used open surgery exclusively to control sepsis in 17 cases: 5 localized collections and 12 anastomotic dehiscences with generalized peritonitis.

Regardless of the approach to septic collections, the bacteriology of the abscess is mandatory to select the antibiotic therapy. Most authors agree that aggressive antibiotic treatment with broad-spectrum antibiotics initiated immediately after the onset of fistula for 7–10 days in patients with local and general symptoms of sepsis, followed by descaling antibiotic therapy according to the antibiogram, decreases mortality by up to 30% [6,9,23,24]. It should be noted that empirical intravenous broad-spectrum antibiotic therapy should not be used routinely in patients with low-output fistulas, no fever, tachycardia, no definite intra-abdominal infection, or no associated wound infection [1,6,9,23,24], as it can lead to the emergence of resistant microbial strains or promote fungal infections. We used broad-spectrum antibiotic therapy from the beginning in 37 (57.81%) cases with definite symptoms of sepsis or with adverse evolution.

Nutritional support (nutrition). Malnutrition, due to inadequate caloric intake by interrupting the oral diet (NPO), hypercatabolism secondary to sepsis, and massive loss of protein, electrolytes, and fluid through fistula is, along with sepsis and hydro-electrolyte imbalances, one of the factors responsible for the failure rate, high morbidity, and mortality [5,6,8,9,10,16,25]. Therefore, nutritional supplementation should be started as soon as possible when the patient has been stabilized. Aggressive nutritional support is used to regain a positive nitrogen balance, maintain the integrity of the intestinal mucosa, reduce fistula output, heal the wound, and reduce the risk of infection [6,8,9,10,11,16,22]. Nutritional status was evaluated dynamically based on clinical (weight, BMI < 18.5, skinfold, arm circumference) and biological (proteinemia, albuminemia, protoalbuminemia, transferrin, and C-reactive protein) criteria. The patient’s nutritional needs were permanently adjusted according to data in the literature: 25–32 kcal/kg/day with a calories/nitrogen ratio of 150:1–200:1, at least 1.5 g/protein/day, to which, depending on the fistula output, vitamin C is added up to 10 times the regular doses, and other vitamins and trace elements (Zn, Cu, Se) at twice the normal doses [8,9,10,22]. In choosing the type of nutrition (parenteral, enteral, or combined) and the route of administration, we considered the fistula topography and flow, digestive tolerance, and general condition of the patient, as well as the logistics and qualifications of the staff.

Total Parenteral Nutrition (TPN), introduced by Duddrick in 1969 [5,25], was one of the significant advances in treating PECF and part of the initial management of the fistulas. Of the patients, 70–80% with high-output fistulas needed, at least initially, TPN [6] shortly after the patient’s stabilization and sepsis control [15]. Nil by mouth (NBM) and TPN reduced the gastrointestinal secretion by 30–50%. They also reduced the fistula output, being useful in hydro-electrolytic rebalancing, reducing the incidence of dehydration and electrolytic imbalances [1,5,10,14,15,24]. Hence, some authors [8] advocate TPN as initial therapy in all cases. The primary disadvantages are the specific complications of the presence of the central venous catheter (sepsis, venous thrombosis, pneumothorax). The controversies are related to insufficient evidence regarding the impact of TPN on spontaneous closure and the morbidity and mortality rates [1,12]. We used it in 16 (25%) high-output fistulas or in patients with digestive intolerance (nausea, abdominal distension, pain) to ensure nutritional intake during the period of complete interruption of oral intake.

Enteral nutrition has gained more ground as the method of choice for all patients with functional gastrointestinal tracts because it preserves the integrity of the intestinal mucosa and the hormonal and immunological function of the intestine [6,9,10]. Most authors agree that, provided there is at least 1.5 m normal bowel proximal or distal to the fistula, and at least 20% of the caloric requirement is administered enterally, the integrity of the mucosa and the hormonal and immunological functions of the bowel are preserved [1,2,10,20,22]. The route of administration differs depending on the location of the fistula; in distal ileal and colic fistulas, food can be administered orally or by nasogastric tube in duodenal fistulas by jejunostomy and in the intermediate ones by fistuloclysis with the feeding tube placed directly in the fistula, under radiological guidance [6].

Enteral nutrition plays an essential role in preventing sepsis by improving protein synthesis, intestinal absorption, and decreasing microbial translocation [2]. It is cheaper and not burdened by the specific complications of parenteral nutrition. Inflammation, strictures, distal obstruction, irradiation, and short bowel are the primary limitations of the method, and the contraindications are represented by the length of the intestine proximal to fistula ˂ 75 cm, discontinuity of the bowel, digestive intolerance, and significant output increase with impairment of the hydro-electrolyte balance. We used it in 19 fistulas (29.68%) with low output and good digestive tolerance. In 29 (43.31%) cases, we used the combined parenteral route in the initial phases of total interruption of oral intake and the enteral route after the patient’s stabilization and sepsis control, with permanent monitoring of fistula output.

Output control is one of the fundamental objectives of non-operative management. It significantly impacts volume, electrolyte balance (especially in patients with high output fistulas), nutrition, skin integrity, and spontaneous fistula closure, although the last has not been proven [1,10,11,26,27]. Output control should be established immediately after fistula identification to prevent fistula loss, skin lesions, local inflammation, pain, and infection [2,8]. Traditionally, the decrease or complete cessation of transit (NBM) and upper digestive aspiration associated with TPN leads to decreased fistula output by reducing intestinal contents and gastrointestinal and biliopancreatic stimulation [10,11,15]. However, the association of nasogastric tube with NPO in the absence of obstruction is not beneficial, causing discomfort and predisposition to complications (gastroesophageal reflux, sinusitis, respiratory complications, etc.) [5,9]. We used the complete cessation of oral intake (NBM) in 16 patients with high-output fistulas. In contrast, in another 35 patients, we opted for the partial decrease of oral intake depending on the evolution of fistula output and their digestive tolerance.

Pharmacological output control includes antacid medication, antisecretory, antiperistaltic drugs, somatostatin, and analogs. Antacid medicines (proton pump inhibitors and histamine receptor blockers) were used in 36 cases for their beneficial effects, reducing fistula output, gastric acid secretion, and the corrosive action of the effluent on the skin; implicitly, it prevents gastritis and stress ulcers [1,2,10,18], even though no data show an increase of spontaneous closure rate [9]. We have no experience with antisecretory or antiperistaltic drugs (loperamide, opium tincture, atropine, codeine), whose beneficial effect in reducing fistula output is cited in the literature [1,21,27]. The use of somatostatin and its synthetic analog (octreotide) is controversial. Somatostatin and its synthetic analogs inhibit the secretion of several gastrointestinal hormones (gastrin, secretin, cholecystokinin, insulin, glucagon, and vasoactive intestinal peptides), inhibit gastric acid and pancreatic secretion, motility, and contractility of the bile duct, reduce the gastrointestinal tract flow, stomach-filling rate, and intestinal motility. Together with TPN and NBM, this led to their use to minimize fistula output [1,5,9,10,11,14,16,26,27]. Adverse effects include intestinal villi atrophy, interruption of bowel adjustment, acute cholecystitis, need for continuous infusion, and costs. Still, the mainly inconclusive and contradictory data regarding the time and rate of fistula closure are arguments for why somatostatin and analogs were not imposed as routine therapy [5,9,10,28]. Our limited experience (only six cases) does not enable any findings on the efficacy and usefulness of this therapy.

Essential actions that must be initiated shortly after identifying the fistula are skin protection and wound care. Skin lesions caused by the corrosive action of the effluent (acid or alkaline depending on the location of the fistula) may appear very early, following 3 h of skin contact, especially in stasis [5,10,21,26]. Chemical irritation caused by intestinal fluid rich in proteolytic enzymes, mechanical trauma due to frequent changes of the collecting bag, allergic reactions to dressings or pouch materials, and infection favored by environmental humidity are the leading causes of skin damage [21,26]. Once occurring, skin lesions (erythema–44 cases, ulcers–12 cases, and necrosis–8 cases in our study) cause pain and discomfort and prevent wound healing. They further limit future management and control options, making it challenging to use collection and containment devices, leading to the digestion of the abdominal wall with the onset of enteroatmospheric fistulas and new lesions of the exposed loops [2,7,10,21,26]. There is a wide range of skincare materials: simple or absorbent dressings, skin barriers (waffles, powders, pastes, sprays, adhesives), collection bags, and suction devices (negative pressure dressings, vacuum-assisted closure system). Their choice must consider the type, location, flow rate of the fistula, and the type of effluent [2,6,7,8,9,10,13,16,26].

We used simple absorbent dressings for low-output fistulas (27 cases) that do not require changing the dressing earlier than 4–5 h. In other cases (high-output fistulas, enteroatmospheric fistulas, or fistulas with uncontrolled flow increasing), we used more advanced techniques, isolated or combined: stoma therapy collection bags (10 cases), ointments and powders (Karaya ointment or zinc), compressive dressings (11 cases) or aspiration-drainage–16 cases. We combined the active suction and the irrigation of the fistulous tract with saline or 4% lactic acid, thus avoiding puddling of the effluent, preventing the contamination of the skin, and reducing the frequency of changing dressings. In the enteroatmospheric fistulas and fistulas with a short tract, we used an original compressive dressing technique, specific to the 1st Clinic of Surgery Craiova: obturation of the external orifice of fistula with a pneumatic elastic balloon of adjustable pressure [29].

We did not use vacuum-assisted closure (VAC) therapy, a state-of-the-art system, which, despite all the benefits, remains controversial and should be used with caution. This is because the negative pressure favors the migration of tissue healing factors and helps heal the wound. VAC is also associated with the increased incidence of new fistulas and a higher mortality rate [15,25]. Regardless of the collection system used, it must allow for patient mobility and be discreet, odor-proof, easy to apply, and remain in place for at least 72 h [2].

Surgery is an essential step in managing PECF and is reserved for solving complications or permanent repair of fistulas that do not close under conservative treatment. Depending on the surgery timing, there are three categories of surgical indications [30]: immediate surgery (within the first 24 h), early surgery (during the next 2–7 days), and delayed reconstructive surgery. The rate of spontaneous fistula closure varies widely (15–75%) according to the makeup of the patient groups and the type of institutional management [5,10]. Therefore, inclusion in one of these categories depends on the morphologic characteristics of the fistula, the type and severity of evolutionary complications, and the conservative therapy response.

Immediate surgery and early surgery are both emergency reinterventions. They include patients with suspected or certain intestinal gangrene, severe peritonitis, life-threatening infection, complete anastomotic dehiscence, and challenging-to-access abdominal collections that are impossible to drain percutaneously using guided CT puncture. Of the reinterventions, 22 of the 26 fell into this category, the surgical indications being generalized peritonitis–12 cases, localized abdominal collections–5 cases, and unfavorable evolution under conservative treatment with a significant increase in flow and worsening of the general condition in another 5 cases. Except for the abscesses in which the reintervention was limited to their open drainage, the resection/reversal of anastomosis with the closure of the distal end and colostomy/ileostomy was the surgery of choice, performed in 9 cases.

Delayed reconstructive surgery was performed only in a limited number of cases (four cases) more than three months after the primary intervention. Two of the cases were persistent fistulas that did not close under conservative treatment. The reintervention consisted of “en bloc” resection of the fistula together with the adjacent intestine in end-to-end anastomosis. The other two cases consisted of repairing the remaining abdominal wall defect after the closure of the enteroatmospheric fistula under conservative treatment.

A unique entity was represented by the fistulas after duodeno-biliopancreatic surgery. Reported to have an incidence of up to 19% in the literature [31], they were encountered in 4 (6.25%) cases: leakage of the hepatico-jejunal anastomosis after cephalic duodenopancreatectomy–3 cases. The other case was a leakage of the hepatic duct and duodenal suture performed for lithiasic acute cholecystitis with biliobiliary and biliary-digestive fistula. The solution adopted was a suture to fix the T-tube drainage of the hepatic duct and separate from the duodenum.

We also recorded two duodenal fistulas after surgery for a duodenal hemorrhagic ulcer (1 case) and after the drainage of a subphrenic abscess after surgery for a hydatid cyst. Reintervention consisted of fistulo-jejunal anastomosis in one case, fistulectomy + suture breach + gastroenteroanastomosis in the other.

The results of our therapeutic strategy were consistent with the data in the literature: 70.31% fistula closure rate, 78.94% after exclusive conservative treatment and 57.61% after surgery, with morbidity of 79.68% and mortality of 29.68%. Further, 21.05% of the fistulas were treated exclusively and conservatively with a 29.68% postoperative mortality.

## 5. Conclusions

PECF management requires a multidisciplinary approach conducted according to an algorithm with well-established objectives and priorities. Resuscitation, sepsis control, output control, skin protection, and nutritional support comprised the initial conservative treatment. Surgery was reserved for complications or permanent repair of fistulas that did not close under conservative treatment.

The therapeutic strategy was adapted to topography, morphological characteristics, fistula output, patient age, general condition, and response to therapy.

## Figures and Tables

**Table 1 medicina-58-00880-t001:** Conservative treatment.

The Objectives of Conservative Treatment	Cases	%
**ICU**		
- Avg. ICU hospitalization (day)–9.93/5.4	35	59.37
- Range–2–22/3–16		
**Resuscitation**	**64**	**100**
- volemic	64	
- hydro-electrolytic	64	
- acid-base	64	
- transfusions	32	50.00
**Sepsis control**	**64**	**100**
- antibiotics	37	57.81
- surgery	17	26.56
**Nutrition:**	**64**	**100**
- total parenteral	16	25.00
- enteral/oral	19	29.68
- both (parenteral + oral/enteral)	29	43.31
**Diminishing fistula output:**		
- NPO	16	25.00
- reducing oral feeding	35	54.68
- H_2_ antagonists	36	56.25
- somatostatine/octreotide	6	9.37
**Skin protection:**	**64**	**100**
- spray, topic, paste	42	65.25
- collection bags	37	59.37
- active suction	16	25.00
- elastic balloon compression	11	17.85
- normal wound dressings (3.2 avg/day)	37	59.37

**Table 2 medicina-58-00880-t002:** Surgery: timing, indications.

Surgery	Cases	%
	26	40.65
**Timing:**		
- <24 h		
- 2–7 days	9	34.61
- >7 days	17	65.39
**Indication**		
- Postoperative peritonitis	12	46.15
- Intraperitoneal abscess	5	19.23
- Evisceration	2	7.69
- Aggravation under conservative treatment	4	15.38
- Raise in output > 1000 mL/24 h	1	3.84
- Removal of a fistulous tract	2	7.69

**Table 3 medicina-58-00880-t003:** Reinterventions: type of surgery.

Primary Surgery		Reoperation		
Lesion	Operation	Intraoperative	Solution	Cases
**Metachronous transverse colon cancer**	Subtotal colectomy + ileum − sigmoid anastomosis	Mesoceliac abscess	Evacuation of pus Drainage	**7**
**Recto-sigmoid junction cancer**	Dixon’s recto-sigmoid resection	Left laterocolic gutter abscess		
**Right colon cancer**	Right colectomy	Right subphrenic abscess		
**Left colon cancer**	Left colectomy	Left laterocolic gutter abscess		
**Inoperable rectal cancer (frozen pelvis)**	Left colostomy	Tumor perforation, pelvic abscess		
**Gastric cancer**	Total gastrectomy	Tumor block, dissection impossible		
**Parastomal hernia after Hartman’s resection**	Anatomic repair of the hernia; segmental enterectomy	Wound abscess: anastomotic dehiscence fixed to the wound		
**Right colon cancer**	Right colectomy	Anastomotic dehiscence	Anastomotic breakdown; closure of colonic stump + ileostomy	**1**
**Sigmoid colon cancer** **Sigmoid volvulus**	Left colectomy Sigmoidectomy	Anastomotic dehiscence	Closure of the distal colon end + left colostomy	**3**
**Transverse colon cancer + recto-colonic polyposis**	Total colectomy + ileum-rectal anastomosis	Ileum-rectum anastomosis dehiscence	Resection of anastomosis; closure of distal end + ileostomy	**1**
**Transverse colon cancer**	Extended right colectomy	Anastomotic leak of the anterior side	Resection of the anastomosis. Closure of the distal end. Ileostomy	**1**
**Peritoneal carcinomatosis after resected small bowel cancer** **Ileostomy closure**	Segmental enterectomy	Tumor perforation Anastomotic leak	Re-entrectomy + end-to-end anastomosis	**2**
**Subhepatic tumor block after resected gastric cancer**	Ileostomy	Ileostomy necrosis	Ileostomy reconstruction	**1**
**Subphrenic abscess after operated liver hydatic cyst**	Partial cystectomy, pericystectomy, drainage	Duodenal leak	Fistulectomy. Suture of the breach. Gastroenteroanastomosis	**1**
**Bleeding duodenal ulcer**	Antrectomy	Duodenal fistula	Roux en Y fistulo-jejunal anastomosis	**1**
**Pancreatic head cancer**	Duodenopancreatectomy	Leak of anterior side of the hepatic-jejunal anastomosis	T-tube drainage of the leak	
**Lithiasis acute cholecystitis + bilio-biliary and bilio-digestive fistula**	Cholecystectomy. Suture of the leak. T-tube drainage	Lesion of the hepatic duct. Breach of the duodenal stump	Suture of the hepatic leak + T-tube drainage	**1**
**Gastric cancer** **Segmental entero-mesenteric infarction**	Total gastrectomy + eso-jejunal anastomosis Segmental enterectomy + end-to-end anastomosis	Right colon lesion Necrosis of ileum and right colon	Right colectomy + ileostomy	**2**
**Recto-colonic polyposis**	Left recto-colectomy	Enteroatmospheric fistula closed conservatively	Cure of evisceration	**2**
**Total reinterventions**				**26**

**Table 4 medicina-58-00880-t004:** Results: evolution, morbidity, mortality.

Evolution	Cases	%
**Closure of fistula**	**45**	**70.31**
- *Conservative treatment exclusively*	*30*	*78.94*
- *Conservative + surgery*	*15*	*57.61*
**Morbidity**	**51**	
***General complications***	** *29* **	**79.68**
- *Sepsis*	*12*	* **45.31** *
- *Cachexia*	*4*	
- *Coagulation disorders*	*1*	
- *Pulmonary*	*3*	
- *Hemorrhagic shock*	*2*	
***Local complications***	** *30* **	
- *Wound suppuration*	*30*	
- *Eviscerations*	*9*	** *46.85* **
- *Acute pancreatitis*	*1*	
**Mortality rate**		
- ***Conservative treatment***	**19**	**29.68**
- ***Conservative treatment + surgery***	** *8* **	** *21.05* **
- ***Cause of death:*** *sepsis 12, MODS 3, severe malnutrition 1, hemorrhagic shock 2, pulmonary thromboembolism 1*	** *11* **	** *29.68* **

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
