# Peer review of "Therapeutic Options in Postoperative Enterocutaneous Fistula—A Retrospective Case Series"

_medicina, 2022, doi:10.3390/medicina58070880_

Round 1

Reviewer 1 Report

In this case series, Maria Mădălina Denicu, et al. analyzed the etiology, risk factors, treatment and outcome of postoperative enterocutaneous fistulas (PECF) in 64 patients and conclude that PECF management requires a multidisciplinary approach conducted according to an algorithm with well-established objectives and priorities.

Strengths of the study:

-        Study question is valid

-       Adequate literature review was performed.

The manuscript can be improved by addressing following concerns.

-        Study objective was not clear. Authors may want to include in this in the introduction section.

-        Describe in detail about the study protocol in the materials and methods section.

-        Authors may want to perform statistical analysis to evaluate what risk factors independently associated with fistula development and closure.

-        Authors may want to look at fistula recurrence rate after surgery and identify variables associated with recurrence.

Author Response

The manuscript can be improved by addressing following concerns.

-        Study objective was not clear. Authors may want to include in this in the introduction section.

-        Describe in detail about the study protocol in the materials and methods section.

-        Authors may want to perform statistical analysis to evaluate what risk factors independently associated with fistula development and closure.

-        Authors may want to look at fistula recurrence rate after surgery and identify variables associated with recurrence”

We performed the following actions:

  • introduced a phrase at the end of the Introduction in which we stated clearly the aim of the paper
  • Introduced a phrase before the last paragraph in “Material and Methods” section, in which we depicted the data we collected from the medical records of the patients
  • The evaluation of the risk factors, independently associated with fistula development and closure made the object of another study published before: “Denicu Maria Mădălina, Chiuțu Cristina Luminița, Șurlin V., Râmboiu S., Ciorbagiu M., Cârțu D, Nemeș R. Etiopathogenic circumstances, anatomical-clinical evaluation, risk factors and prediction in enterocutaneous postoperative fistulas Journal of Surgery [Jurnalul de chirurgie]. 2021; 17(4): 250 –261, [Article in Romanian] DOI: 10.7438/JSURG.2021.04.04
  • We introduced a phrase at the end of the “Results” section in which we stated that “we didn’t registered any recurrences after the closure of the fistula by conservative or conservative+surgery treatment”

Reviewer 2 Report

The paper is interesting for two reasons:

1.       It is presentation of the personal experience of the authors.

2.       In the discussion there is an extensive review of the International literature.

I think that in the discussion, the authors might to write more to explain the relatively high rate of morbidity and mortality in their serie of patients.

Author Response

Thank you! Regarding the high rates of morbidity and mortality, the study was conducted in an Emergency County Hospital. So most of the patients were presenting in the Emergency Room with different known or, most of them, unknown co-morbidities. 

Reviewer 3 Report

This is an excellent report of one of the most vexing complications in surgery.I would kindly suggest that the authors review the text for minor spelling, sentence structure, and syntax.

Author Response

“….kindly suggest that the authors review the text for minor spelling, sentence structure, and syntax”

Our manuscript was edited by a specialized service in editing according to the certificate attached to the present letter. However, we checked again the manuscript for spelling and synthax.

Round 2

Reviewer 1 Report

Authors addressed all my concerns. I have no issues publishing this manuscript.